# The Novel Phosphatidylinositol-3-Kinase (PI3K) Inhibitor Alpelisib Effectively Inhibits Growth of PTEN-Haploinsufficient Lipoma Cells

**DOI:** 10.3390/cancers11101586

**Published:** 2019-10-17

**Authors:** Anna S. Kirstein, Adrien Augustin, Melanie Penke, Michele Cea, Antje Körner, Wieland Kiess, Antje Garten

**Affiliations:** 1Pediatric Research Center, University Hospital for Children and Adolescents, Leipzig University, 04103 Leipzig, Germany; anna.kirstein@medizin.uni-leipzig.de (A.S.K.); adrien.augustin@student.uliege.be (A.A.); melanie.penke@medizin.uni-leipzig.de (M.P.); antje.koerner@medizin.uni-leipzig.de (A.K.); wieland.kiess@medizin.uni-leipzig.de (W.K.); 2Faculty of Medicine, University of Liège, 4000 Liege, Belgium; 3Chair of Hematology, Department of Internal Medicine (DiMI), University of Genoa, 16100 Genoa, Italy; michele.cea@unige.it; 4IRCCS Polyclinic Hospital San Martino, 16100 Genoa, Italy; 5Institute of Metabolism and Systems Research, University of Birmingham, Birmingham B15 2TT, UK

**Keywords:** mTOR, AKT, PHTS, PROS, lipoma, rapamycin, overgrowth, proliferation, spheroids, ribosomal protein S6

## Abstract

Germline mutations in the tumor suppressor gene *PTEN* cause PTEN Hamartoma Tumor Syndrome (PHTS). Pediatric patients with PHTS frequently develop lipomas. Treatment attempts with the mTORC1 inhibitor rapamycin were unable to reverse lipoma growth. Recently, lipomas associated with PIK3CA-related overgrowth syndrome were successfully treated with the novel PI3K inhibitor alpelisib. Here, we tested whether alpelisib has growth-restrictive effects and induces cell death in lipoma cells. We used PTEN-haploinsufficient lipoma cells from three patients and treated them with alpelisib alone or in combination with rapamycin. We tested the effect of alpelisib on viability, proliferation, cell death, induction of senescence, adipocyte differentiation, and signaling at 1–100 µM alpelisib. Alpelisib alone or in combination with rapamycin reduced proliferation in a concentration- and time-dependent manner. No cell death but an induction of senescence was detected after alpelisib incubation for 72 h. Alpelisib treatment led to a reduced phosphorylation of AKT, mTOR, and ribosomal protein S6. Rapamycin treatment alone led to increased AKT phosphorylation. This effect could be reversed by combining rapamycin with alpelisib. Alpelisib reduced the size of lipoma spheroids by attenuating adipocyte differentiation. Since alpelisib was well tolerated in first clinical trials, this drug alone or in combination with rapamycin is a potential new treatment option for PHTS-related adipose tissue overgrowth.

## 1. Introduction

Patients with germline mutations in the tumor suppressor gene *PTEN* show a wide variety of phenotypes related to cellular overgrowth. There are several syndromes associated with *PTEN* mutations, including Cowden syndrome, Proteus syndrome, and Bannayan–Riley–Ruvalcaba syndrome (BRRS), all subsumed under the term PTEN Hamartoma Tumor Syndrome (PHTS) [1]. Symptoms include an increased risk for cancer (breast, endometrial, thyroid), macrocephaly, vascular malformations, polyps of the gastrointestinal tract and other hamartomas, and, especially in the BRRS type, early-onset lipoma development [2]. Lipomatosis in pediatric patients can be life-threatening, as the infiltrating growth of lipomatous masses can obstruct vital organ function and can lead to chronic pain conditions. In some patients, resection as the only current treatment option is impossible due to lipoma position or poor general condition of the patient. Treatment attempts with the mechanistic target of rapamycin complex 1 (mTORC1) inhibitor rapamycin were shown to improve the general condition of PHTS patients [3,4], but could not reverse lipoma growth [4]. PTEN antagonizes the phosphoinositide-3-kinase (PI3K)/AKT/mTOR signaling pathway which regulates cellular metabolism and promotes cellular growth, proliferation, and survival [5]. PI3K lies downstream of several growth factor receptors and upon activation catalyzes the reaction of phosphatidylinositol (4,5)-bisphosphate (PIP_2_) to phosphatidylinositol (3,4,5)-trisphosphate (PIP_3_). PIP_3_ is the key molecule to activate further downstream signaling components, e.g., the pro-survival molecule AKT. PTEN acts as a lipid phosphatase on PIP_3_, catalyzing the conversion to PIP_2_, and therefore is a negative regulator of the AKT/mTOR signaling cascade [6]. mTORC1 regulates AKT activity through a negative feedback loop via its target ribosomal protein S6 kinase. An inhibition of mTORC1 by rapamycin leads to an increased activation of AKT [7]. This loss of negative feedback inhibition of AKT might be a cause for the reduced efficacy of rapamycin observed in a treatment attempt of a child with PHTS-associated lipoma [4].

Recently, patients with lipomatous tumors associated with a related spectrum of syndromes caused by mosaic activating PI3K mutations (PIK3CA-related overgrowth syndrome, PROS) were successfully treated with the novel PI3K inhibitor alpelisib (BYL-719) [8]. The size of patients’ tumors was reduced after few months and side effects were reported to be manageable. Alpelisib is a selective PI3Kα inhibitor designed for the use in human cancer therapy [9]. It was tested in several clinical trials alone or in combination with other chemotherapeutics against solid tumors [10,11,12]. Here, we tested proliferation, induction of apoptosis, and signaling pathway activation in two-dimensional (2D) and three-dimensional (3D) cultures of PTEN-haploinsufficient primary lipoma cells treated with alpelisib. We aimed to determine whether alpelisib has growth-restrictive effects and would induce cell death in lipoma cell cultures from pediatric patients with PHTS.

## 2. Results

### 2.1. Effect of Alpelisib on Proliferation of Lipoma Cells

#### 2.1.1. Alpelisib Reduced Cell Viability in a Dose- and Time-Dependent Manner

We treated five different primary lipoma cell cultures with alpelisib concentrations ranging from 1 to 50 µM and measured cell viability (the number of metabolically active cells) using the WST-1 assay after 72 h for alpelisib alone (Figure 1a) or in combination with 10 nM rapamycin (Figure 1b). Additionally, we tested cell viability at different time points (24–144 h) in LipPD1 cells for alpelisib alone (Figure 1c) and in combination with rapamycin (Figure 1d).

We noted a concentration-dependent decrease in cell viability for all cell cultures (*p* < 0.0001). At 10 µM alpelisib, cell viability was significantly decreased in all cell cultures. A combination of 10 µM alpelisib and 10 nM rapamycin further decreased cell viability compared to single treatment (*p* < 0.001). IC_50_ values for 72 h WST-1 assays were calculated to be 9.09 µM (LipPD1), 6.31 µM (LipPD2), 18.33 µM (LipPD3), 6.7 µM (Lip3), and 15.74 µM (Lip4). A summary of all IC_50_ values is given in Appendix A. We found similar effects for PHTS (LipPD1-3) and PROS patients’ lipoma cells (Lip3 and Lip4). During six days of treatment, cell viability decreased in a concentration- (*p* < 0.0001) and time-dependent manner (*p* < 0.0001). We observed a further decrease in cell viability for a combined treatment of alpelisib and rapamycin. To examine whether combining alpelisib and rapamycin has synergistic or additive anti-lipoma activity, LipPD1, LipPD2, and LipPD3 cells were treated with alpelisib and rapamycin across a range of concentrations. Analysis of the combined effect by the Chou and Talalay method [13] showed a significant decrease in viability after co-treatment compared to either agent alone, with a combination index of <1.0 in all tested cells. Overall, these data confirmed the synergistic anti-lipoma activity of this drug combination (Appendix A).

#### 2.1.2. Alpelisib Inhibited Lipoma Cell Proliferation

To analyze whether the effects seen in the WST-1 assays are caused by an inhibition of proliferation, we counted Hoechst-stained nuclei of lipoma cells after alpelisib, rapamycin, or combined treatment (Figure 2a–d) and performed immunofluorescence staining of the proliferation marker Ki-67 in LipPD1 cells after 48 h alpelisib treatment (Figure 2e–f).

Alpelisib attenuated growth of all three lipoma cell cultures alone and in combination with rapamycin. For LipPD1 cells treated with 100 µM alpelisib, cell count was stable for six days in culture. The same observation was made for a combination of 10 µM alpelisib and 10 nM rapamycin. IC_50_ values for 72 h Hoechst assays were calculated to be 15.91 µM (LipPD1), 10.06 µM (LipPD2), and 15.79 µM (LipPD3) (Appendix A). The fraction of Ki-67 positive cells was reduced after alpelisib treatment for 1 µM to 0.75 ± 0.07 fold (*p* = 0,074), for 10 µM to 0.55 ± 0.06 fold (*p* = 0.018), and for 100 µM to 0.22 ± 0.1 fold (*p* = 0.017) in a concentration-dependent manner (*p* = 0.0098). Ki-67 immunofluorescence staining in LipPD2 and LipPD3 cells after 72 h also showed a reduced fraction of proliferation marker positive cells (Appendix A).

### 2.2. Cytotoxicity of Alpelisib in Lipoma Cells

We measured apoptosis induction in lipoma cells after 72 h alpelisib treatment by annexin V/PI staining via flow cytometry (Figure 3a), and cell death after 24 h and 72 h via LDH-cytotoxicity assay (Figure 3b).

We did not observe cell death after 72 h 50 µM alpelisib treatment in LipPD1 and Lip3 cells. The fraction of viable cells was slightly reduced in Lip4 cells after 50 µM alpelisib treatment compared to solvent control. The fraction of dead and apoptotic cells was highly elevated in the positive control (50 µM perifosine). For LipPD1 cells, we did not observe apoptosis after 72 h 10 nM rapamycin treatment alone, while the total fraction of apoptotic and dead cells was slightly increased (by 9.6 ± 2.3%, *p* = 0.0471) after a combined treatment with alpelisib and rapamycin. To confirm our findings, we performed LDH cytotoxicity assays for LipPD1, LipPD2, and LipPD3 cells after 24 h and 72 h of 50 µM alpelisib treatment (Figure 3b). We did not observe any additional LDH release in 50 µM alpelisib treated lipoma cells, implying no induction of cell death. We neither observed cytotoxicity for 10 nM rapamycin nor a combination of 50 µM alpelisib with 10 nM rapamycin.

### 2.3. Effect of Alpelisib on PI3K Signaling

To determine whether alpelisib affects downstream PI3K signaling, we performed Western blot analyses, qPCRs, and immunofluorescence staining of alpelisib-treated LipPD1 cells to detect activated signaling components.

#### 2.3.1. Alpelisib Reduced PI3K/AKT/mTOR Pathway Activation in Lipoma Cells

To determine basal pathway activation, cells were treated with alpelisib for 24 h and AKT, mTOR, and ribosomal protein S6 phosphorylation was determined via Western blot analysis (Figure 4a–c). pS6-immunofluorescence staining of LipPD1 cells was performed after 48 h alpelisib treatment (Figure 4d–e).

AKT activation was reduced in 50 µM alpelisib-treated cells (*p* = 0.019), while rapamycin enhanced AKT phosphorylation (*p* = 0.192). This effect of rapamycin was repressed when combined with alpelisib (*p* = 0.279). Activation of mTOR was reduced in 10 µM (*p* = 0.099) and 50 µM (*p* = 0.066) alpelisib-treated cells, as well as in 10 nM rapamycin-treated cells (*p* = 0.18). Phosphorylation of S6 was significantly reduced for all tested alpelisib and rapamycin concentrations. Western blots for LipPD2 (Appendix A) and LipPD3 (Appendix A) cells showed similar results. All blots, GAPDH loading control blots and densitometric analyses are provided in Appendix A. In immunofluorescence staining experiments, the fraction of pS6 positive cells was reduced after alpelisib treatment for 10 µM to 0.57 ± 0.02 fold (*p* = 0.0074) and for 100 µM to 0.01 ± 0.006 fold (*p* = 0.0169). pS6 immunofluorescence staining in LipPD2 and LipPD3 cells after 72 h showed similar results (Appendix A).

#### 2.3.2. Alpelisib Reduced *PCNA*, *GLUT1,* and *PGK* mRNA Expression

To determine the influence of alpelisib on gene expression, we performed reverse transcription quantitative PCR (RT-qPCR) analysis for the proliferation marker *PCNA*, the glucose transporter *GLUT1,* and the glycolysis enzyme *PGK* (Figure 5). All three genes are known to be regulated by PI3K signaling [14,15,16].

*PCNA* mRNA was downregulated after 24 h alpelisib treatment (to 0.67 ± 0.08 fold). Similarly, we observed a downregulation of *GLUT1* mRNA (to 0.60 ± 0.11 fold) and *PGK* mRNA (to 0.67 ± 0.06 fold).

### 2.4. Effect of Alpelisib on Adipocyte Differentiation in 2D and 3D Models

To test whether alpelisib would attenuate adipogenesis, we differentiated LipPD1 cells in 2D culture and 3D spheroid models to assess lipid accumulation (Figure 6a–b), differentiation markers (2D) (Figure 6c–e), and spheroid size (3D) (Figure 6f–g).

We observed reduced lipid accumulation after alpelisib treatment of LipPD1 cells in adipocyte differentiation medium for 10 days. The fraction of adipocytes was reduced from 54.8 ± 7% to 29.9 ± 7%. Further, we investigated the mRNA expression of differentiation markers with or without 10 µM alpelisib in adipogenic medium and detected *peroxisome proliferator-activated receptor γ* (*PPARγ*) mRNA downregulated after 10 days in differentiation medium with 10 µM alpelisib to 0.55 ± 0.06 fold. *Adiponectin* mRNA was downregulated to 0.54 ± 0.04 fold and *adipocyte protein 2* (*aP2*) mRNA to 0.54 ± 0.03 fold. *Fatty acid synthase* (*FASN*) mRNA was downregulated as well to 0.58 ± 0.09 fold, *p* = 0.064 (Appendix A). To confirm our findings, we evaluated the effect of alpelisib treatment on adipocyte differentiation in a 3D spheroid model [17]. While the size of control 3D-spheroids increased during in vitro differentiation (to 1.25 ± 0.06 fold after 10 days), the size of 10 µM alpelisib-treated spheroids was reduced after 4 days (to 0.78 ± 0.05 fold) and, afterwards, stable throughout the duration of the experiment (to 0.79 ± 0.05 fold after 10 days). The differences between controls and alpelisib treated cells were significant from day 4 on (*p* = 0.0063 at day 4).

### 2.5. Effect of Alpelisib on Senescence of Lipoma Cells

To check whether the lipoma cells maintained their mesenchymal stem cell phenotype or underwent senescence during alpelisib treatment, we performed a β-galactosidase senescence staining (Figure 7a–b) and measured gene expression of stem cell and senescence markers (Figure 7c–e).

The fraction of senescent cells after 72 h alpelisib treatment was elevated 2.71 ± 0.47 fold, indicating cell cycle arrest. Similar observations were made for LipPD2 (2.43 ± 0.7 fold, *p* = 0.236) and LipPD3 cells (4.96 ± 1.98 fold, *p* = 0.065) (Appendix A). Since we observed an inhibition of adipogenesis through alpelisib, we investigated whether these undifferentiated cells would retain their mesenchymal stem cell state or undergo senescence by analyzing the mRNA expression of senescence marker *CDKN2A* (*p16*) and stem cell markers *CD44* and *Thy-1* (*CD90*) with or without 10 µM alpelisib in adipogenic medium. *p16* mRNA was upregulated after 10 days in differentiation medium with 10 µM alpelisib (to 5.4 ± 2.05 fold). *CD44* mRNA was upregulated to 1.96 ± 0.32 fold. In contrast, *CD90* mRNA was reduced to 0.71 ± 0.05 fold. Similar trends in the regulation of *adiponectin*, *p16*, *CD44,* and *CD90* were observed in LipPD2 and LipPD3 cells (Appendix A).

## 3. Discussion

Lipomatosis associated with germline mutations in *PTEN* can be a severe complication in pediatric patients. So far, the only treatment option is surgical resection, which is not feasible in every patient and has to be repeated because of lipoma recurrence. In this research work, we tested effects of the novel PI3K inhibitor alpelisib on PTEN-haploinsufficient lipoma cells (stromal-vascular fraction) from pediatric patients. Proliferation of these preadipocyte-like cells is enhanced, potentially due to a constitutive overactivation of the PI3K-pathway [18]. The PI3K/AKT/mTOR axis acts as a major growth factor signaling pathway regulating proliferation and survival [19]. In adipose tissue, PI3K signaling is essential for adipocyte differentiation [20]. We asked whether alpelisib influences these cellular responses and, by doing so, can inhibit growth of lipoma cells.

When we treated lipoma cells with alpelisib, we observed a concentration-dependent decrease in cell viability. This decrease was in line with a reduced cell number after alpelisib treatment. The fraction of proliferation marker Ki-67 positive cells was decreased in a concentration-dependent manner and the proliferation marker PCNA was downregulated on the mRNA level. While these findings indicate that alpelisib negatively influences cell proliferation of PTEN-haploinsufficient lipoma cells, apoptosis could not be induced during 72 h alpelisib treatment. Indeed, we did not detect any alpelisib-induced cytotoxicity in any of the cell lines tested. The number of dead/apoptotic cells was also only slightly elevated when we combined the mTOR inhibitor rapamycin with alpelisib. Nevertheless, this combination was more effective than treatment with the same concentrations of either drug alone in terms of inhibition of proliferation. Supporting this finding, we detected a synergistic effect of rapamycin and alpelisib treatment on cell viability. In line with the findings of Keam et al., in head and neck cancer cell lines, we detected a reduction of proliferation using alpelisib at similar concentrations [21].

In the majority of cancers with mutated PI3K, the PI3Kα isoform is activated [22]. Alpelisib is a selective PI3Kα inhibitor [9]. As hypothesized, alpelisib inhibited phosphorylation and, therefore, activation of AKT in the lipoma cells. This also affected the downstream PI3K-pathway molecule ribosomal protein S6, which is phosphorylated by the ribosomal protein S6 kinase beta 1 (S6K1). S6K1 is known to be a key molecule in transcriptional regulation of cell cycle and cell size [23]. Alpelisib inhibited S6 phosphorylation to the same extent as direct mTOR inhibition through rapamycin. We found that after 48 h treatment with 100 µM alpelisib, only approximately 1% of cells showed S6 phosphorylation. When treating lipoma cells with rapamycin, we observed an increased phosphorylation of AKT, confirming earlier findings in lipoma cells [4]. This AKT activation induced by inhibition of the negative feedback regulation by mTORC1 [7] could be reversed when combining rapamycin with 10 µM alpelisib. This indicates that resistance mechanisms occurring during rapamycin treatment of PHTS patients might be overcome when co-treating with alpelisib. Moreover, an inhibition of further upstream molecules of the PI3K pathway through alpelisib may overcome the limitations of targeting mTOR. PTEN directly antagonizes PI3K, and in patients with germline *PTEN* mutations, this leads to an activation of the whole pathway. PI3K inhibitors limit this pathway activation on the same level as PTEN, which might provide advantages over an inhibition further downstream.

Many cancer cells are known to rely on aerobic glycolysis as their primary source of energy generation. The PI3K pathway plays a major role in regulating this metabolic process [24]. To further dissect molecular causes for the observed growth arrest after alpelisib treatment, we quantified gene expression of the glucose transporter *GLUT1* and the glycolysis enzyme *PGK*. Both were significantly reduced, indicating changes in glucose metabolism on the transcriptional level during alpelisib treatment, along with a reduction of proliferating cell nuclear antigen (*PCNA)* expression, a marker for proliferating cells. In patients, this might be advantageous to mainly target cells with a transformed metabolic profile, and therefore selectively inhibit malignancies.

To test alpelisib in a physiological model of lipoma adipocyte differentiation, we used 3D cultures of lipoma cells. Size of lipoma cell spheroids in 3D culture increased during 10 days in differentiation medium, as shown before [17], which is probably due to a cell volume increase because of lipid accumulation. This effect was not observed when treating lipoma spheroids with 10 µM alpelisib, which, in contrast, led to a reduced spheroid size. This could result both from an inhibition of differentiation and from a reduction of cell size through downregulation of S6 phosphorylation [25]. In PHTS patients, a reduction in lipoma size without surgery could counteract disease progression, and therefore prevent life-threatening complications. mTOR is known to regulate adipocyte differentiation through PPARγ, the master transcriptional regulator of adipogenesis [26]. Similar to rapamycin [27], we found that alpelisib reduced adipocyte differentiation in 2D lipoma cells culture as well. The differentiation markers *PPARγ*, *adiponectin*, *aP2,* and *FASN* were downregulated.

We finally tested whether the attenuated adipocyte differentiation observed in alpelisib-treated lipoma cells was caused by preservation of stem cell characteristics. When checking gene expression of CD44 and CD90, markers present on adipose-derived mesenchymal stem cells, we found gene expression differences in differentiated lipoma cells that were alpelisib-treated compared to controls. We found that the stem cell marker CD90 was downregulated. CD90 downregulation is associated with senescence in mesenchymal stem cells [28]. In contrast, CD44 was upregulated. Although considered a mesenchymal stem cell marker [29], CD44 was also found to be a senescence-induced cell adhesion gene [30]. Supporting these data, we detected upregulation of *CDKN2A* (*p16*) by alpelisib, both in differentiated and undifferentiated lipoma cells. Moreover, alpelisib enhanced β-galactosidase staining of lipoma cells. These findings indicate that alpelisib induced senescence both in PTEN haploinsufficient preadipocytes and differentiated adipocytes. In a mouse model for PROS syndrome, no induction of senescence or apoptosis was detected [8].

Alpelisib was successfully used to treat overgrowth of different tissues associated with activating *PI3K* mutations in PROS patients [8]. Since activating PI3K mutations and deactivating PTEN mutations both activate the PI3K-pathway, we hypothesize beneficial effects of alpelisib for PHTS patients as well. In our cell culture experiments, we indeed observed similar effects on cell viability in PROS and PHTS patients’ lipoma cells. We found growth-inhibitory effects with PI3K-pathway inhibition and reduction in size of in vitro lipoma 3D models. The main adverse event found in clinical trials with alpelisib was hyperglycemia [10,11,12]. Alpelisib was tested in a set of pediatric patients with PROS where it was well tolerated, and side effects were reported to be mild and manageable [8]. We therefore consider alpelisib as a potential future alternative to rapamycin treatment or as an option for combination therapy with rapamycin for patients with PHTS-associated lipoma. As a next step, we propose testing of alpelisib in vivo. We will test effects of alpelisib on PTEN knockout and wildtype tissues in a conditional PTEN-knock-out mouse model and investigate the reversibility of these effects.

## 4. Materials and Methods

### 4.1. Cell Culture and Adipocyte Differentiation

We used cells of the stromal vascular fraction isolated from lipomas of three different pediatric PHTS and two PROS patients (Leipzig University ethical approval: no. 425-12-171220). Mutations were detected by Sanger sequencing [18], while the PTEN deletion seen in LipPD1 was analyzed by multiplex ligation-dependent probe amplification (MLPA) and array-based comparative genomic hybridization (Array-CGH) [4]. A reduction in PTEN mRNA and/or protein, as well as increased AKT phosphorylation, compared to PTEN wildtype controls was found [18]. Isolation and culture methods were described previously [4,18]. Table 1 contains a list of the lipoma cells used.

For differentiation, we plated 13,000 lipoma cells/cm² in culture medium and changed medium to differentiation medium (DMEM/F12 containing 2 µmol/L rosiglitazone, 25 nmol/L dexamethasone, 0.5 mmol/L methylisobuthylxantine, 0.1 µmol/L cortisol, 0.01 mg/mL apotransferrin, 0.2 nmol/L triiodotyronin, and 20 nmol/L human insulin [31]) after 24 h. Cells were incubated with 10 µM alpelisib (Selleckchem, Munich, Germany) or solvent control (DMSO) in differentiation medium for 10 days. The medium was replaced every 72 h. For lipid staining, cells were fixed in Roti-Histofix 4% (Carl Roth GmbH, Karlsruhe, Germany), washed with DPBS, and stained with Oil Red O solution (0.03% in 60% isopropanol, Sigma, St. Louis, MO, USA) for 15 min at 37 °C [32]. For transcriptional comparison to undifferentiated cells, these were kept in serum-free culture medium for 4 days.

A modified method according to Klingelhutz et al. was used for scaffold-free 3D cultures of lipoma cells [17]. To form spheroids, 10,000 cells per well were seeded into low attachment 96-well microplates (PS, U-bottom, clear, cellstar^®^, cell-repellent surface, Greiner Bio-One, Kremsmünster, Austria) in 100 µl differentiation medium and incubated for 24 h. After 24 h, spheroids were incubated with 10 µM alpelisib or solvent control in differentiation medium for 10 days. Half of the medium was replaced every 72 h. Microscope images were taken daily using the EVOS FL Auto 2 Cell Imaging System (Invitrogen; Thermo Fisher Scientifc, Inc., Waltham, MA, USA). Image analysis to determine the spheroid size was performed using ImageJ [33].

### 4.2. Cell Viability/Proliferation

For proliferation assays, cells were seeded at a density of 3000 cells/well (24 h) or 1500 cells/well (48–144 h) on 96-well plates and treated with different concentrations of alpelisib and/or rapamycin (both Selleckchem, Munich, Germany). Medium was replaced every 72 h. Cell viability measures were performed using the Cell Proliferation Reagent WST-1 (Roche Diagnostics GmbH, Mannheim, Germany) according to the manufacturer’s protocol. After WST-1 assay, cells were fixed and nuclei were stained with Hoechst 33342 (Sigma, Munich, Germany) for 5 min at a concentration of 10 µg/ml in DPBS. Hoechst fluorescence was detected at 455 nm.

### 4.3. Apotosis Assay

For apoptosis detection, cells were seeded at a density of 1750 cells/cm² and treated with alpelisib/alpelisib+rapamycin for 72 h. Then, 50 µM perifosine (Selleckchem, Munich, Germany) was used as a positive control. The FITC-Annexin V Apoptosis Detection Kit I (BD Pharmingen, Franklin Lakes, NJ, USA) was used for annexinV–fluorescein isothiocyanate and propidium iodide staining of the trypsinized cells. Apoptosis was detected by flow cytometry.

### 4.4. LDH Cytotoxcicity Assay

For LDH cytotoxicity assays, cells were seeded at a density of 3000 cells/well on 96-well plates in culture medium. Cells were treated for 72 h with 50 µM alpelisib, 10 nM rapamycin, or a combination of both. LDH release was detected using the CyQUANT™ LDH Cytotoxicity Assay (Invitrogen; Thermo Fisher Scientifc, Inc., Waltham, MA, USA) according to the manufacturer’s protocol.

### 4.5. Western Blot Analysis

For Western blot analysis, lipoma cells were seeded at a density of 10,000 cells/cm² in culture medium. Cells were incubated with different alpelisib and/or rapamycin concentrations for 24 h. Proteins were extracted and immunoblotting was performed as described elsewhere [4]. We used 10 µg protein per lane and incubated with primary antibodies (Cell Signaling Technology, Inc., Danvers, MA, USA) and secondary antibodies (Dako; Agilent Technologies, Inc., Santa Clara, CA, USA), according to Table 2. Glyceraldehyde-3-phosphate dehydrogenase (GAPDH) (Merck KGaA, Darmstadt, Germany) was used as loading control.

### 4.6. Immunofluorescence Staining

For immunofluorescence staining, cells were fixed in 4% PFA after 48 h of alpelisib treatment. Cells were permeabilized and blocked in IF-buffer (DPBS + 5% BSA + 0.3% Tween20) for 1 h at RT and stained with pS6 or Ki-67 primary antibodies over night at 4 °C (Table 2). Cells were washed three times with IF-buffer and incubated with secondary antibodies (Invitrogen; Thermo Fisher Scientifc, Inc., Waltham, MA, USA) for 2 h at RT in the dark. We took microscope images using the EVOS FL Auto 2 Cell Imaging System (Invitrogen; Thermo Fisher Scientific, Inc., Waltham, MA, USA). Counting of nuclei (DAPI channel) and pS6/Ki-67 positive cells (GFP channel) was performed using the Celleste Image Analysis Software (Thermo Fisher Scientific).

### 4.7. Reverse Transcription Quantitative PCR (RT-qPCR)

For transcriptional analysis, we seeded 5000 cells/cm² in culture medium and extracted mRNA after 24 h of treatment with 10 µM alpelisib or solvent control. For transcriptional analysis on differentiation, cells were seeded at a density of 13,000 cells/cm². Cells were kept in differentiation medium with or without 10 µM alpelisib for 10 days before RNA was extracted. RNA extraction, reverse transcription, and qPCR were performed as previously described [18]. Table 3 contains a list of primers used for qPCR assays (Applied Biosystems; Thermo Fisher Scientifc, Inc.). *Glucose transporter 1* (*GLUT1*), *phosphoglycerate kinase* (*PGK*), and *Proliferating cell nuclear antigen* (*PCNA*) mRNA was normalized to the housekeeper *Tata box binding protein* (*TBP*) mRNA. *PPARγ*, *adiponectin*, *aP2*, *FASN*, *p16*, *CD44,* and *CD90* mRNA were normalized to the housekeepers *β-actin* or *HPRT* mRNA.

### 4.8. Senescence β-Galactosidase Staining

To detect senescence, we seeded 2000 cells/well on 96-well plates in culture medium. After 72 h of treatment with 10 µM alpelisib or solvent control, we performed fixation and senescent cell staining using the Senescence β-Galactosidase Staining Kit (Cell Signaling Technology, Inc., Danvers, MA, USA) according to the manufacturer’s protocol. We used 60 µl β-galactosidase staining solution per well and after overnight incubation, we added 50 µL Hoechst in DPBS (1 ng/mL) and took microscope images using the EVOS FL Auto 2 Cell Imaging System (pictures of 5% of the well area). We counted nuclei (DAPI channel) with the Celleste Image Analysis Software (Thermo Fisher Scientific) and β-galactosidase positive cells (blue, phase contrast) using ImageJ [33].

### 4.9. Statistical Analysis

Means of at least three independent experiments were statistically analyzed using GraphPad Prism 6 software (GraphPad Software, Inc., La Jolla, CA, USA). For comparison of two conditions, means of independent experiments were compared via unpaired Student’s *t*-test. For multiple comparisons, we used one- or two-way analysis of variance (ANOVA), followed by a post-hoc Tukey’s multiple comparisons test (one-way ANOVA) or Dunnett’s multiple comparisons test (two-way ANOVA). To determine the significance of fold changes, we used one-sample *t*-tests of the log(fold change) and compared to the hypothetical value 0 [34]. To calculate IC_50_ values, we used GraphPad Prism 6 (nonlinear fit, log(inhibitor) vs. response, least squares fit, standard slope model). Isobologram analysis [13] was done using the CalcuSyn software program.

## 5. Conclusions

Lipomatosis in pediatric PHTS patients is a rare condition, but nevertheless, can affect individuals in a life-threatening manner due to limited treatment options. In this research project, we tested the PI3K inhibitor alpelisib on PTEN-haploinsufficient lipoma cells from PHTS patients. We found that alpelisib inhibited cellular growth and reduced the size of 3D lipoma cell cultures, although alpelisib did not induce apoptosis. In contrast to rapamycin, alpelisib inhibited activation of AKT, promising to overcome drug resistance mechanisms occurring during rapamycin treatment in PHTS patients [4].

## Figures and Tables

**Figure 1 cancers-11-01586-f001:**
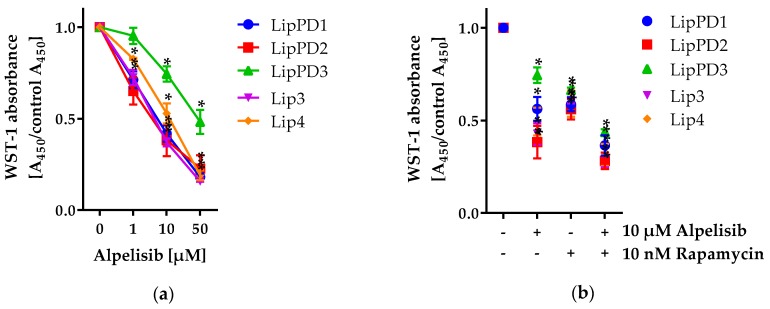
WST-1 cell viability assay after alpelisib or alpelisib+rapamycin treatment of lipoma cell cultures from three PTEN Hamartoma Tumor Syndrome (PHTS) patients (LipPD1-3) and two PIK3CA-related overgrowth spectrum (PROS) patients (Lip3-4): (**a**) 72 h alpelisib treatment reduced cell viability at concentrations ≥10 µM in all tested cell cultures; (**b**) a 72 h combined treatment with alpelisib and rapamycin further decreased cell viability; (**c**) 24–144 h treatment of PTEN-haploinsufficient LipPD1 cells at 1 to 100 µM alpelisib reduced cell viability; (**d**) 24–144 h combined treatment with alpelisib and/or rapamycin further decreased viability. Fold over solvent control (black line), *n* = 3, * *p* ≤ 0.05.

**Figure 2 cancers-11-01586-f002:**
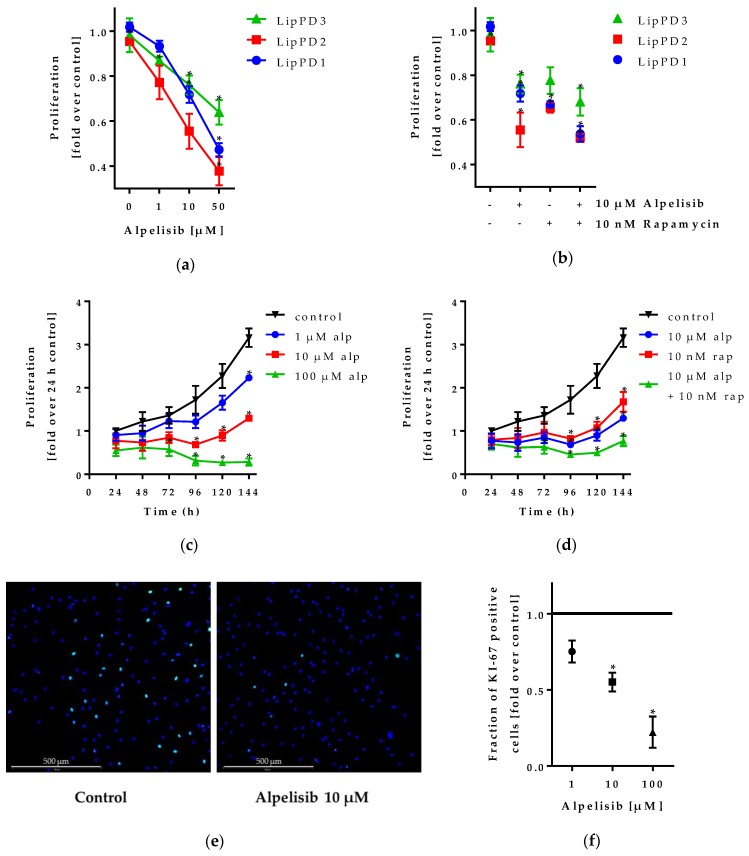
Hoechst proliferation assay and Ki-67 proliferation marker immunofluorescence staining after alpelisib/alpelisib+rapamycin treatment in PTEN-haploinsufficient lipoma cell cultures: (**a**) 72 h alpelisib treatment attenuated proliferation in a concentration-dependent manner in three different lipoma cell cultures; (**b**) a combined treatment with alpelisib and rapamycin further decreased cell count. Fold over solvent control, *n* = 3, * *p* ≤ 0.05. (**c**) Cell counts in LipPD1 cells during six days of 100 µM alpelisib (alp) treatment did not increase, while cells in solvent control proliferated; (**d**) a combination of rapamycin (rap) and alpelisib further decreased proliferation compared to treatment with alpelisib or rapamycin alone. *n* ≥ 3, * *p* ≤ 0.05 compared to solvent control. (**e**) Ki-67 proliferation marker immunofluorescence staining of LipPD1 cells after 48 h of alpelisib treatment: Merged image of nuclei (blue) and Ki-67 (green) for untreated and 10 µM alpelisib treated cells; (**f**) fraction of Ki-67 positive cells is decreased after alpelisib treatment, fold over solvent control (black line), *n* = 3, * *p* ≤ 0.05.

**Figure 3 cancers-11-01586-f003:**
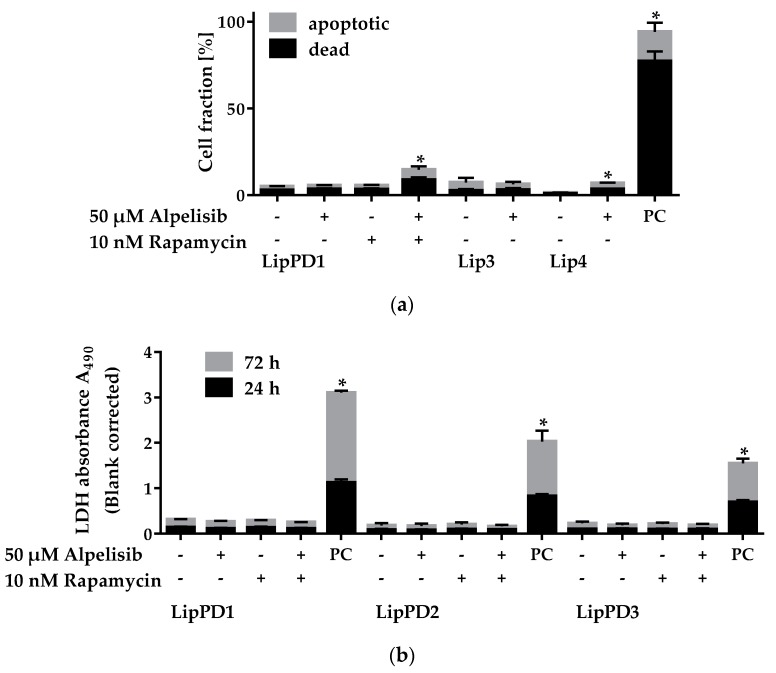
Apoptosis/cell death of PTEN Hamartoma Tumor Syndrome (PHTS, LipPD1-3) and PIK3CA-related overgrowth spectrum (PROS, Lip3-4) patients’ lipoma cells after 50 µM alpelisib and/or 10 nM rapamycin treatment. (**a**) Apoptotic/dead cells determined via annexin V/PI staining did not increase compared to solvent control in LipPD1 and Lip3 cells, while 50 µM perifosine (PC, positive control) led to cell death. A minor rise in the fraction of apoptotic/dead cells was observed for Lip4 and for a combination with 10 nM rapamycin in LipPD1 cells. (**b**) LDH cytotoxicity assay in LipPD1-3 cells showed no increase in cell death after 24 h or 72 h 50 µM alpelisib and/or 10 nM rapamycin treatment. PC: Cell lysis positive control, *n* = 3, * *p* ≤ 0.05.

**Figure 4 cancers-11-01586-f004:**
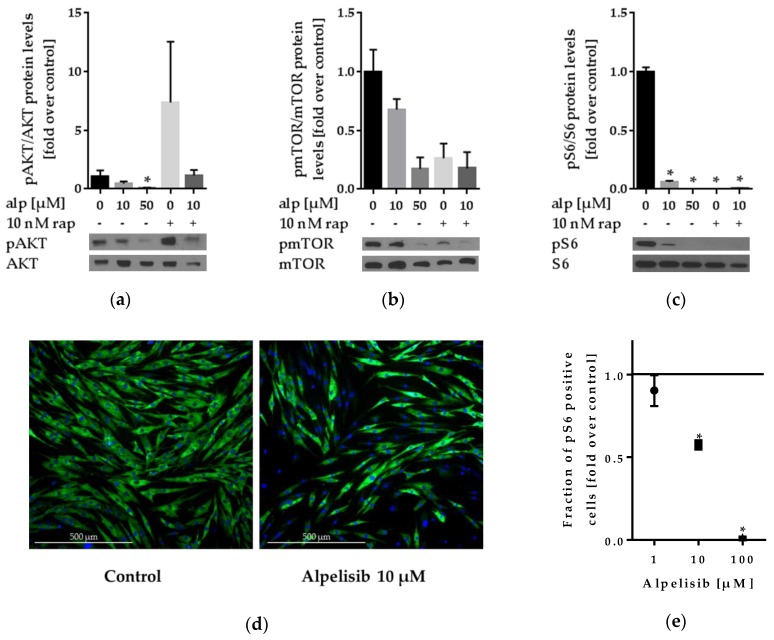
Western blot analysis of LipPD1 cells after 24 h treatment and pS6 immunofluorescence staining after 48 h treatment with alpelisib (alp) and/or rapamycin (rap), representative blots and densitometric analysis: (**a**) Phosphorylated AKT (phospho-Thr 308 (pAKT) normalized to total AKT) was reduced after alpelisib but not rapamycin treatment; (**b**) phosphorylation of mTOR (phospho-Ser 2448 (pmTOR) normalized to total mTOR); (**c**) phosphorylation of ribosomal protein S6 (phospho-Ser 235/236 (pS6) normalized to total S6 protein) was reduced after alpelisib and rapamycin treatment. One representative blot out of three for each protein (phosphorylated and total) and densitometric analysis of three independent experiments is shown, fold over solvent control, * *p* ≤ 0.05. (**d**) Merged image of nuclei (blue) and pS6 (green) immunofluorescence staining for untreated and 10 µM alpelisib-treated cells; (**e**) fraction of pS6 positive cells decreased after alpelisib treatment, fold over solvent control (black line), *n* = 3, * *p* ≤ 0.05.

**Figure 5 cancers-11-01586-f005:**
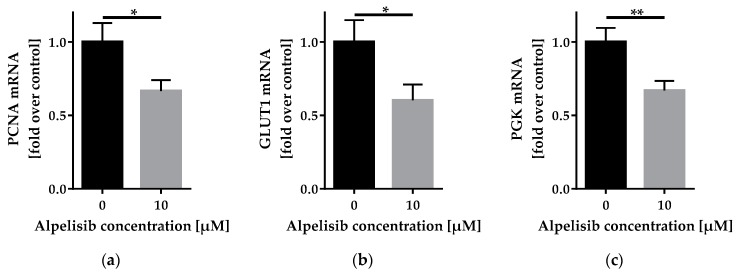
RT-qPCR of cell cycle and glucose metabolism genes from LipPD1 cells after 24 h treatment with 10 µM alpelisib: mRNA expression of (**a**) *PCNA*, (**b**) *GLUT1,* and (**c**) *PGK* was downregulated in 10 µM alpelisib-treated cells. Fold over control, values were normalized to the housekeeper gene *TBP*, *n* = 6, * *p* ≤ 0.05, ** *p* ≤ 0.01.

**Figure 6 cancers-11-01586-f006:**
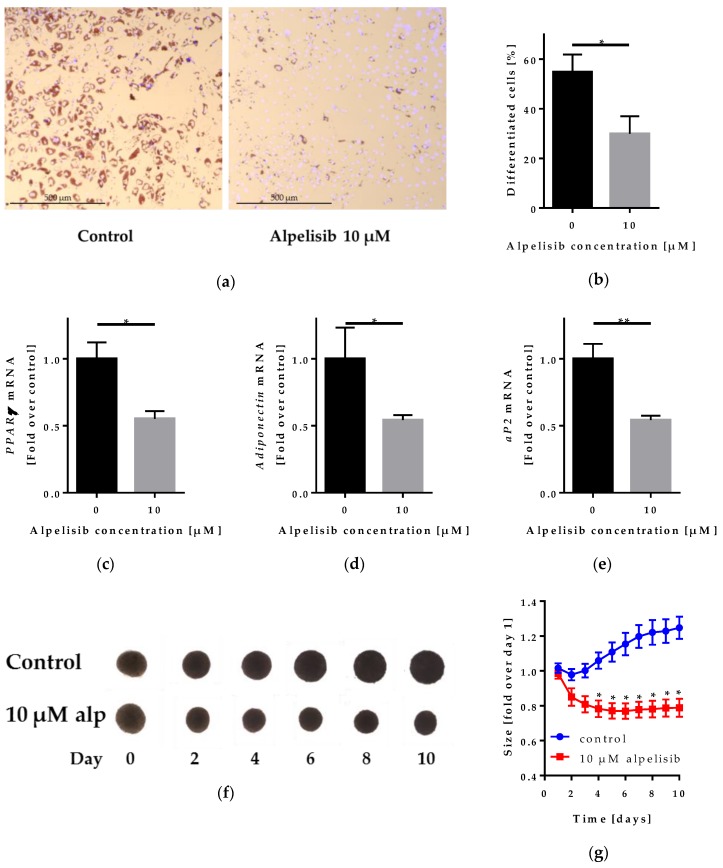
Alpelisib treatment of LipPD1 cells during adipocyte differentiation: (**a**) Oil Red O lipid (red) and Hoechst (blue) staining after 10 days in differentiation medium showed reduced lipid accumulation after 10 µM alpelisib treatment; (**b**) the percentage of adipocytes after 10 days in differentiation medium was decreased with 10 µM alpelisib (alp) compared to solvent control, *n* = 3, * *p* = 0.016. RT-qPCR of adipogenesis-related genes in LipPD1 cells after 10 days of differentiation with or without 10 µM alpelisib: mRNA expression of (**c**) *PPARγ*, (**d**) *Adiponectin,* and (**e**) *aP2* was downregulated in 10 µM alpelisib-treated cells. Fold over control, values were normalized to the housekeeper gene *β-actin*, *n* = 3, * *p* ≤ 0.05, ** *p* ≤ 0.01. (**f**) 10 µM alpelisib treatment of 3D LipPD1 cell cultures during adipocyte differentiation; (**e**) size of 3D lipoma spheroids during 10 days in differentiation medium decreased with 10 µM alpelisib (alp) compared to day 1, values for day 1 were calculated by normalizing to the mean of day 1; for following days, each spheroid size was divided by the size of the same spheroid at day 1, *n* = 4, * *p* ≤ 0.05.

**Figure 7 cancers-11-01586-f007:**
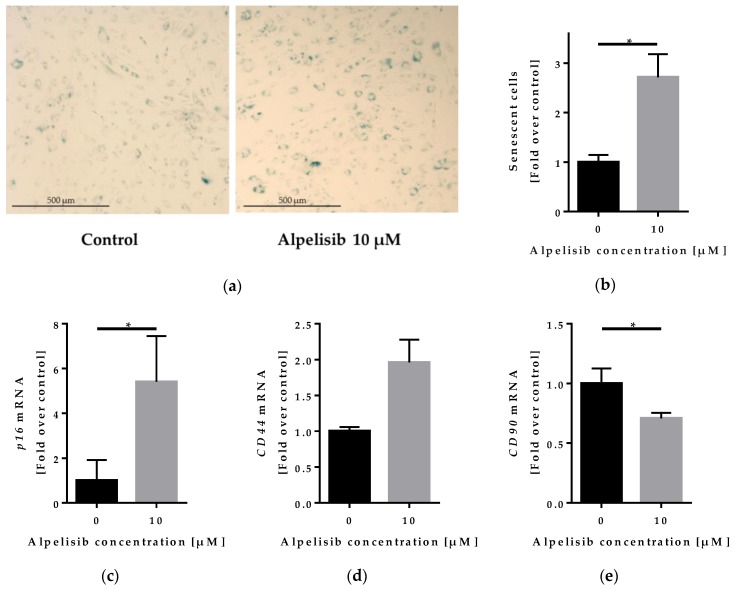
β-galactosidase senescence staining of LipPD1 cells after 72 h alpelisib treatment: (**a**) β-galactosidase staining showed a higher number of senescent (blue) cells after 10 µM alpelisib treatment; (**b**) the fraction of senescent cells increased with 10 µM alpelisib compared to solvent control, *n* = 3, * *p* = 0.027. RT-qPCR of senescence and stem cell marker genes from LipPD1 cells after 10 days of differentiation with or without 10 µM alpelisib: mRNA expression of (**c**) *p16* was upregulated, (**d**) *CD44* was upregulated (*p* = 0.053), and (**e**) *CD90* was downregulated in 10 µM alpelisib-treated cells. Fold over control, values were normalized to the housekeeper gene *β-actin*, *n* = 3, * *p* ≤ 0.05.

**Table 1 cancers-11-01586-t001:** Lipoma cells from pediatric PHTS and PROS patients [18].

Name	Sex	Age at Resection	Mutation
LipPD1	male	3	Heterozygous deletion of *PTEN* exons 2-9 of 9
LipPD2	male	3	*PTEN* heterozygous point mutation (c.404T>A, p.I135K)
LipPD3	female	4	*PTEN* heterozygous point mutation (exon 1, c.76A>C, p.T26P)
Lip3	female	14	Mosaic *PI3KCA* mutation (10%) (exon 10, c.1624G>A, p.(Glu542Lys)
Lip4	male	13	Mosaic *PI3KCA* mutation (30-40 %) (exon 8, c.1340_1366del, p.(Pro447_Leu455del)

**Table 2 cancers-11-01586-t002:** Antibodies used for Western blot (Wb) and immunofluorescence staining (IF).

Primary Antibody	Dilution	Supplier	Cat. no
mTOR (7C10) Rabbit mAb	1:1000 TBS-T 5%BSA (Wb)	CST	#2983
Phospho-mTOR (Ser2448) Rabbit mAb (D9C2)	1:1000 TBS-T 5%BSA (Wb)	CST	#5536
AKT antibody Rabbit polyclonal Ab	1:1000 TBS-T 5%BSA (Wb)	CST	#9272
Phospho-AKT (Thr308) (224F9) Rabbit mAb	1:500 TBS-T 5%BSA (Wb)	CST	#4056
S6 ribosomal protein (5G10) Rabbit mAb	1:1000 TBS-T 5%BSA (Wb)	CST	#2217
Phospho-S6 Ribosomal protein (Ser235/236) (D57.2.2E) XP^®^ Rabbit mAb	1:1000 TBS-T 5%BSA (Wb) 1:500 IF-buffer (IF)	CST	#4858
GAPDH (6C5) Mouse mAb	1:50,000 TBS-T 5% milk (Wb)	Merck	MAB374
KI-67 (MIB-1) Mouse mAb	1:200 IF-buffer (IF)	Dako	P0447
**Secondary antibody**	**Dilution**	**Supplier**	**Cat. no**
Polyclonal goat anti-rabbit immunoglobulin/HRP	1:2000 TBS-T 5% milk (Wb)	Dako	P0447
Polyclonal goat anti-mouse immunoglobulin/HRP	1:2000 TBS-T 5% milk (Wb)	Dako	P0448
Alexa Fluor 488 goat anti-mouse IgG H+L	1:1000 IF-buffer (IF)	Invitrogen	A11001
Alexa Fluor 488 goat anti-rabbit IgG H+L	1:1000 IF-buffer (IF)	Invitrogen	A11034

**Table 3 cancers-11-01586-t003:** Primers used for RT-qPCR.

Gene	Forward	Reverse	Probe
*TBP*	TTGTAAACTTGACCTAAAGACCATTGC	TTCGTGGCTCTCTTATCCTCATG	AACGCCGAATATAATCCCAAGCGGTTTG
*β-actin*	CGA GCG CGG CTA CAG CTT	TGC TTG TGT TGG GTG GAT ATT G	TGG CCA CCG ACT CCT ACA AGG TTA CTC AC
*HPRT*	GGC AGT ATA ATC CAA AGA TGG TCA A	GTC TGG CTT ATA TCC AAC ACT TCG T	CAA GCT TGC TGG TGA AAA GGA CCC C
*GLUT1*	TGGCATCAACGCTGTCTTCT	AGCCAATGGTGGCATACACA	
*PGK*	GAATGGGAAGCTTTTGCCCG	GCAGTGTCTCCACCACCTATG	
*PCNA*	CTAAAATGCGCCGGCAATGA	TCTCCTGGTTTGGTGCTTCA	
*Adiponectin*	GGC CGT GAT GGC AGA GAT	CCT TCA GCC CCG GGT ACT	CGATGTCTCCCTTAGGACCAATAAGACCTGG
*aP2*	GCTTTTGTAGGTACCTGGAAACTTG	ACA CTG ATG ATC ATG TTA GGT TTG G	CCTGGTGGCAAAGCCCACTCCTCAT
*PPARγ*	GATCCAGTGGTTGCAGATTACAA	GAGGGAGTTGGAAGGCTCTTC	TGACCTGAAACTTCAAGAGTACCAAAGTGCAA
*FASN*	GGCAAATTCGACCTTTCTCAGA	GGACCCCGTGGAATGTCA	CACCCGCTCGGCATGGCTATCTT
*p16*	CTTCGGCTGACTGGCTGG	TCATCATGACCTGGATCGGC	
*CD44*	Predesigned assay (ID: Hs01075864_m1)		
*CD90*	Predesigned assay (ID: Hs00264235_s1)

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
