# Peer review of "The Novel Phosphatidylinositol-3-Kinase (PI3K) Inhibitor Alpelisib Effectively Inhibits Growth of PTEN-Haploinsufficient Lipoma Cells"

_cancers, 2019, doi:10.3390/cancers11101586_

Round 1

Reviewer 1 Report

Dear Editor,

Followings are my second review comments to the authors of “The novel phosphatidylinositol-3-kinase (PI3K) inhibitor alpelisib effectively inhibits growth of PTEN-haploinsufficient lipoma cells. (Cancers-602086)

In this revised manuscript, the authors performed some experiments and added new data such as Fig. 10, 12 and 13, and presented that alpelisib suppresses adipocyte differentiation of PTEN-haploinsufficient lipoma cells. They also showed that alpelisib treatment induced senescence marker: b-Gal as well as expression of p16 and CD44, and decreased CD90 expression. Based on these results, the authors concluded that alpelisib induced cell senescence in PTEN-haploinsufficient lipoma cells, but not adipocyte differentiation and stem cell induction. Although these findings are very intriguing, the findings of Fig. 10, 12 and 13 were inconsistent with those shown in Fig. 1, 2, and 3. Because of the inconsistencies between them, the manuscript lacks logical consistency and needs sufficient interpretation based on scientific evidence.

In general, terminally differentiated cells enter senescent state showing atrophic change, halting metabolism and cell proliferation, and finally resulting in apoptosis. The results of Fig. 4 (decreased cell proliferation), Fig. 8 (downregulated metabolism) and Fig. 11 (induced atrophy) nicely provide the evidence of cell senescence of PTEN-haplosufficient lipoma cells treated by alpelisib.

However, the authors described that cell death (or apoptosis) was not observed in PTEN-haplosufficient lipoma cells by 50 uM alpelisib treatment after 72 h. In reply to reviewer, the authors explained the results of cell viability assay are caused by an inhibition of cell proliferation. If cell proliferation was suppressed by alpelisib treatment, how the authors interpret the results of Fig. 1and 2 (decreased cell viability)? The authors should clarify what phenomena was induced on PTEN-haploinsufficient lipoma cells by alpelisib treatment.

The authors need to perform further experiments or should plausible interpretation that is consistent with all of these results.

Author Response

Comment: In this revised manuscript, the authors performed some experiments and added new data such as Fig. 10, 12 and 13, and presented that alpelisib suppresses adipocyte differentiation of PTEN-haploinsufficient lipoma cells. They also showed that alpelisib treatment induced senescence marker: b-Gal as well as expression of p16 and CD44, and decreased CD90 expression. Based on these results, the authors concluded that alpelisib induced cell senescence in PTEN-haploinsufficient lipoma cells, but not adipocyte differentiation and stem cell induction. Although these findings are very intriguing, the findings of Fig. 10, 12 and 13 were inconsistent with those shown in Fig. 1, 2, and 3. Because of the inconsistencies between them, the manuscript lacks logical consistency and needs sufficient interpretation based on scientific evidence.

In general, terminally differentiated cells enter senescent state showing atrophic change, halting metabolism and cell proliferation, and finally resulting in apoptosis. The results of Fig. 4 (decreased cell proliferation), Fig. 8 (downregulated metabolism) and Fig. 11 (induced atrophy) nicely provide the evidence of cell senescence of PTEN-haplosufficient lipoma cells treated by alpelisib.

However, the authors described that cell death (or apoptosis) was not observed in PTEN-haplosufficient lipoma cells by 50 uM alpelisib treatment after 72 h. In reply to reviewer, the authors explained the results of cell viability assay are caused by an inhibition of cell proliferation. If cell proliferation was suppressed by alpelisib treatment, how the authors interpret the results of Fig. 1and 2 (decreased cell viability)? The authors should clarify what phenomena was induced on PTEN-haploinsufficient lipoma cells by alpelisib treatment.

The authors need to perform further experiments or should plausible interpretation that is consistent with all of these results.

Reply: Thank you for your comment, we however do not see any inconsistency in measuring a reduction in WST-1 assay values (Fig. 1 and 2) and in cellular proliferation (Fig. 3) as well as observing the induction of senescence (Fig. 12 and 13) with increasing alpelisib concentrations. The measurements of the WST-1 assay (also termed cell viability assay) indicate the activity of mitochondrial dehydrogenases, which can be altered by changes in proliferation, cell death or metabolic activity. The results from WST-1 assays represent the number of metabolically active cells in each well. This is measured by a change in absorption of the WST-1 reagent caused by reduction of the soluble formazan reagent employed in the assay. A reduced proliferation would therefore correlate with reduced absorbance. To make this clearer, we changed the term “cell viability” into “WST-1 absorbance” for figure 1 a-d. For the inhibition of differentiation by alpelisib we want to point out that the PI3K pathway is involved in both cellular proliferation as well as differentiation depending upon cellular environment. By providing cells with differentiation medium we trigger adipogenesis, but signaling seems to be blocked by alpelisib treatment. We therefore do not see a need for further experiments.

Reviewer 2 Report

The authors had attempted to answer all my concerns and comments.

Author Response

Thank you for your comment.

Reviewer 3 Report

The manuscript of AS Kirstein et al aims to investigate the effect of alpelisib/ BYL719 -a novel selective PI3K-alpha selective inhibitor- on the growth of PTEN haplo-insufficient lipoma cells. PTEN is a tumor suppressor that inhibits PI3K/AKT pathway through its lipid phosphatase activity, but also exerts phosphatase-independent biological actions, e.g. TP53 stabilization, DNA damage repair. The heterozygous germline mutation of PTEN is associated with the development of syndromes characterized by the development of harmatomas and lipomas - benign tumors that may be life-threatening-, and increased risks of cancers. The manuscript is in line with a publication from another group showing that alpelisib efficiently improved the disease symptoms of CLOVES syndrome in mouse model and in patients. CLOVES syndrome is characterized by congenital lipomatous overgrowth as a result of post-zygotic mosaic gain-of-function mutations in the PIK3CA gene.

The manuscript of AS Kirstein et al extends these observations and reports that alpelisib decreased the growth in vitro and the adipocyte differentiation of PTEN-deficient lipoma cells, and cooperated with the mTOR inhibitor rapamycin. Alpelisib biological effect was not associated with induction of apoptosis or cell death but triggered cell senescence.

One major interest of this study relies on the use of primary culture of lipoma cells from three patients with PTEN harmatoma tumor syndrome.

Comments:

The manuscript requires a reorganization of the illustrations. This would markedly improve the message. It is uncomfortable to have to switch to these too many small Figures in the main text and in the supplementary section. Figure 1/S1 and Fig2 should be gathered; same comments for Fig3, Fig 4, Fig S3; for Fig 5 and Fig S4; for Fig6, Fig7 & S3b; for Fig8, Fig9, Fig10, Fig11; for Fig 12 & 13;   …

WST-1 assay allows to evaluate the activity of mitochondrial deshydrogenase. Thus, changes in absorbance at 440nm -that reflects the amount of formazan produced- might be related to changes in cell proliferation and/or cell death. The term "Cell viability (fold over control)" for the y-axis is not really accurate, since it corresponds to OD450nm. At least, change "fold over control" to "% of control".

This not clear why in Fig 1, in absence of alpelisib the curves concerning LipD2 and LIPD3 are below 100% viability, since 100% corresponds to untreated cells (solvent control).

More points could have been performed to establish IC50 (50% inhibition) values, especially lower doses and higher doses to reach a plateau. Which formula was used to calculate IC50 ? The horizontal bar (100%) should be removed from the Figure.

Same comments for Fig3. It seems difficult to provide an IC50 value in absence of plateau.

The SD or SEM values of the IC50 should be added in table S1.

In Fig1b, align symbols with the corresponding treatments.

From what I understood, the time-course of alpelisib (Figure 2) represents for each time point the relative OD440nm of alpelisib treated cells standardized with the corresponding untreated cells.  This representation leads to a loss of information concerning cell proliferation. Since cell seeding was performed at higher cell density for T24, T24 and T72 should not be joined by a line.

Figure 3d, Please, indicate the statistical test used. A Tukey posthoc test should allow to delineate any difference between the different treatments.

Figure 4 and Figure S2a. Are the different length of treatment (48h vs 72h) related to the different sensitivity of LipD1 and LipD2/ LipD3 cells to alpelisib ?

It would have been also interesting to include in some experiments, (e.g. proliferation assay, apoptosis, differentiation) a culture of wild-type human preadipocytes to evaluate the impact of PTEN deficiency in these processes.

In addition to the evaluation of the accumulation of senescence marker by qPCR in differentiated LIPD cells (Fig 13), It would have been interesting to perform concurrently a B-galactosidase senescence staining to compare their relative sensitivity to alpelisib with their undifferentiated counterparts (Fig 12).

Line 253 change CD40 to CD44

One interesting point that might be later on addressed concerns the recovery of cell proliferation after cell treatment with alpelisib. In the absence of in vivo models this might provide interesting input concerning the reversibility of alpelisib treatment on senescence/ proliferation/ differentiation to control tissue overgrowth and resistance during long term exposure.

Author Response

Comment: The manuscript requires a reorganization of the illustrations. This would markedly improve the message. It is uncomfortable to have to switch to these too many small Figures in the main text and in the supplementary section. Figure 1/S1 and Fig2 should be gathered; same comments for Fig3, Fig 4, Fig S3; for Fig 5 and Fig S4; for Fig6, Fig7 & S3b; for Fig8, Fig9, Fig10, Fig11; for Fig 12 & 13;   …

Reply: Thank you for your suggestions, we changed the figures accordingly except for the following:

We did not include data from Fig. S3a into Fig. 4 and Fig. S3b into Fig. 7 because measurements were taken at different time points. We did not perform alpelisib treatment on LipPD2 and 3 cells for 48 h due to time and resource limitations. Fig 8 was kept separately because gene expression measurements were performed using undifferentiated cells, while assays relating to adipocyte differentiation were performed in cells cultured in differentiation medium.

Additionally we combined Fig 6 and 7 to Fig. 4.

Comment: WST-1 assay allows to evaluate the activity of mitochondrial deshydrogenase. Thus, changes in absorbance at 440nm -that reflects the amount of formazan produced- might be related to changes in cell proliferation and/or cell death. The term "Cell viability (fold over control)" for the y-axis is not really accurate, since it corresponds to OD450nm. At least, change "fold over control" to "% of control".

Reply: Following this suggestion, we changed the y-axis labeling into “WST-1 absorbance [A450/control A450]”.

Comment: This not clear why in Fig 1, in absence of alpelisib the curves concerning LipD2 and LIPD3 are below 100% viability, since 100% corresponds to untreated cells (solvent control).

Reply: We agree with the reviewer. The solvent control (DMSO, used to normalize measured values) differs from the untreated cells in culture medium. Since this seems to be a technical problem, we omitted the untreated cell values.

Comment: More points could have been performed to establish IC50 (50% inhibition) values, especially lower doses and higher doses to reach a plateau. Which formula was used to calculate IC50 ? The horizontal bar (100%) should be removed from the Figure.

Same comments for Fig3. It seems difficult to provide an IC50 value in absence of plateau.

The SD or SEM values of the IC50 should be added in table S1.

Reply: We agree with the reviewer that more points would improve IC50 calculations. We could not use higher amounts of alpelisib due to solubility limitations. We calculated IC50 values using the GraphPad Prism 6 software (log(inhibitor) vs. response,  standard slope model) and added this information to the methods section (line 427-428). As suggested, we added SEM values to table S1.

Comment: In Fig1b, align symbols with the corresponding treatments.

Reply: We made the suggested changes. 

Comment: From what I understood, the time-course of alpelisib (Figure 2) represents for each time point the relative OD440nm of alpelisib treated cells standardized with the corresponding untreated cells.  This representation leads to a loss of information concerning cell proliferation. Since cell seeding was performed at higher cell density for T24, T24 and T72 should not be joined by a line.

Reply: We agree with the reviewer about the loss of information on proliferation in this specific graph, but we provided proliferation data of the same cells in Figure 3c. We made the suggested changes in Figure 1 d.

Comment: Figure 3d, Please, indicate the statistical test used. A Tukey posthoc test should allow to delineate any difference between the different treatments.

Reply: As described in section 4.9. Statistical analysis, we performed a two-way ANOVA followed by a post hoc Dunnett's multiple comparisons test which is recommended by GraphPad Prism. We used Tukey's multiple comparisons tests for data compared by one-way ANOVA. 

Comment: Figure 4 and Figure S2a. Are the different length of treatment (48h vs 72h) related to the different sensitivity of LipD1 and LipD2/ LipD3 cells to alpelisib ?

Reply: We did not perform alpelisib treatment on LipPD2 and 3 cells for 48 h due to time and resource limitations, this is why we provide data for 72 h.

Comment: It would have been also interesting to include in some experiments, (e.g. proliferation assay, apoptosis, differentiation) a culture of wild-type human preadipocytes to evaluate the impact of PTEN deficiency in these processes.

Reply: We agree that this is of scientific interest, but not relevant for PHTS patients, since they have a germline PTEN mutation. We however added this point to the outlook section (line 329).

Comment: In addition to the evaluation of the accumulation of senescence marker by qPCR in differentiated LIPD cells (Fig 13), It would have been interesting to perform concurrently a B-galactosidase senescence staining to compare their relative sensitivity to alpelisib with their undifferentiated counterparts (Fig 12).

Reply: We first performed beta-galactosidase staining on differentiated cells, but found all cells stained blue (in controls and treated). This is why we decided to work on undifferentiated cells. We assume beta-galactosidase staining is not applicable for adipocytes, since brief literature search did not reveal any previous studies using this method in adipocytes.

Comment: Line 253 change CD40 to CD44

Reply: Changed as suggested.

Comment: One interesting point that might be later on addressed concerns the recovery of cell proliferation after cell treatment with alpelisib. In the absence of in vivo models this might provide interesting input concerning the reversibility of alpelisib treatment on senescence/ proliferation/ differentiation to control tissue overgrowth and resistance during long term exposure.

Reply: We thank the reviewer for this interesting suggestion and added this point to the outlook section (line 330). We plan a mouse study to validate cell culture findings.

This manuscript is a resubmission of an earlier submission. The following is a list of the peer review reports and author responses from that submission.

Round 1

Reviewer 1 Report

Pediatric patients with PTEN Hamartoma Tumor Syndrome (PHTS) frequently develop lipomas. In this study, Kirstein AS et al. examined whether PI3K inhibitor alpelisib has growth-restrictive effects and induces apoptosis in lipoma cells, since mTORC1 inhibitor rapamycin has shown with no significant effect on reversing lipoma growth. The authors used PTEN-haploinsufficient lipoma cells from three patients and treated them with alpelisib alone or in combination with rapamycin, and showed that 1-100 µM alpelisib alone or in combination with 10 nM rapamycin reduced the cell proliferation in a concentration- and time-dependent manner. The phosphorylation levels of AKT, mTOR and ribosomal protein S6 were reduced in alpelisib-treated cells. Rapamycin treatment alone led to increased AKT phosphorylation, while alpelisib could reduce rapamycin-induced AKT phosphorylation levels, the viability and proliferation of lipoma cells. Since alpelisib has been well tolerated in first clinical trials, this drug may exhibit a potential to be a therapeutic compound for lipoma cells. However, there are several issues needed to be clarified.

1.      In Figure 1, the authors should calculate the IC50 values of Alpelisib on the lipoma cells from three different patients. Is there an additional or synergistic effect of Alpelisib in combination with Rapamycin treatment on lipoma cell viability? In addition to PTEN-haploinsufficient lipoma cells, what is the effect of Alpelisib on wild-type PTEN lipoma cells?

2.      In Figure 2 and 3, what are the IC50 values of Alpelisib on these three lipoma cell viability and proliferation? Is it possible that Alpelisib has an effect on inducing the apoptosis of these three lipoma cells? Since there is a big inhibitory effect of Alpelisib on the cell viability after the 24-hour treatment, is this big effect of Alpelisib on the cell viability due to growth suppression, apoptosis induction or both?

3.      In Figure 4, 48 h, 10 μM Alpelisib treatment can reduce the cell proliferation marker Ki-67 in lipoma cells. How about the dose-response or time-kinetic effect of Alpelisib on the cell proliferation of the other two lipoma cells?

4.      In Figure 5, please show the whole results from the examination of the effects of Alpelisib on the cell death and apoptosis of these three lipoma cells. What are the IC50 values of Alpelisib on these three lipoma cell death and apoptosis?

5.      In Figure 6, please explain why rapamycin can induce the phosphorylation levels of AKT in LipPD1 cells. Is this phenomenon also shown in LipPD2 and LipPD3 cells? The results should be further statistically calculated to see whether there is a statistical significance between groups.

6.      In Figure 7, whether the phenomenon of Alpelisib-suppressed pS6 positive cells is also observed in the other two lipoma cells?

Author Response

Dear reviewer

Thank you for the careful evaluation of our manuscript. Attached please find our point-by-point response to your comments

Reviewer 2 Report

Dear Editor,

Followings are my review comments to the authors of “The novel phosphatidylinositol-3-kinase (PI3K) inhibitor alpelisib effectively inhibits growth of PTEN-haploinsufficient lipoma cells. (Cancers-537425)

In this manuscript, the authors presented that novel PI3K inhibitor, alpelisib alone or in combination with rapamycin reduced proliferation of PTEN-haploinsufficient lipoma cells. They also described that alpelisib treatment reduced phosphorylation of AKT, mTOR and ribosomal protein S6. Although these findings are intriguing and worth investigating, the manuscript demands sufficient evidence to support authors’ conclusions. Furthermore, there are following concerns about the study and the authors should pay attention to the interpretation of the results.

The review comments to the authors are as follows.

The proportion of apoptotic cells after 72 h treatment with alpelisib and rapamycin showed slight elevation, but the authors did not observe cell death after 72 h treatment. However, alpelisib alone or combination with rapamycin treatment after 72 h reduced both cell viability and proliferation. The authors should clarify the causes of cell viability and proliferation reduction, especially 72 h after treatment.

In adipocyte differentiation in 2D and 3D models, the authors showed that alpelisib treatment attenuated adipocyte differentiation and decreased the size of 3D lipoma spheroids during 10 days. There is no explanation in the manuscript what phenomena was induced on LipPD1 cells by alpelisib treatment. When adipocyte differentiation was inhibited by alpelisib, did LipPD1 cells keep immature state such as mesenchymal stem cell or induce cell senescence? The authors should perform further experiments to reveal what type of cellular reaction was induced by alpelisib.

The authors analyzed the effects of alpelisib with the concentration ranges from 1mM to 100mM. How the authors determine these concentration ranges? The data of cytotoxicity test of alpelisib against LipPD1 cells should be demonstrated.

In Western blotting, an internal control such as b-actin or GAPDH should be included.

Author Response

(The authors gave the same response as above.)

Reviewer 3 Report

Overall, this is a straight forward study of the effects of a PI3-K inhibitor, alpelisib (alp), in suppressing viability and proliferation of PTEN-haploinsufficient lipoma, either alone or in combination with mTOR inhibitor, rapamycin. 

Please refer to the following comments when revising the manuscript:

Table 1. Can the authors discuss the status of the remaining PTEN allele? Especially for LipPD1. Can PTEN protein be detected to qualify the use of the term haploinsufficient lipoma? How can the authors be sure that the remaining “wild-type” alleles are not inactivated through methylation, promoter deletion, etc? It will be very informative if a Western blot including all three LipPD cell lysates analyzed for PTEN, and other signaling molecules, if feasible. 

Fig. 5 Only minimal apoptotic cell death (<10%) in alp treated LipPD1 cells. Have the authors investigated other mechanisms of cell death such as autophagy, or necroptosis?

Fig. 6 and 7.  There appears to be some discrepancy between the >90% reduction in pS6/S6 levels in LipPD1 cells at 10 uM alp and the mere 50% drop in the fraction of pS6 positive cells by IF.

Author Response

(The authors gave the same response as above.)
